# Mediation and moderation analysis of the association between physical capability and quality of life among stroke patients

**Xue Yang**[ID]°, **Hongmei Zhang**°, **Qian Liu**, **Yihuan Lu**, **Liqing Yao**[ID]*

Department of Rehabilitation Medicine, the Second Affiliated Hospital of Kunming Medical University, Kunming, Yunnan, China

° These authors contributed equally to this work.

* yaoliqing98731@163.com

## Abstract

To estimate the extent to which physical capability, depressive symptoms, balanced self-efficacy (BSE), and other risk factors, that are interrelated with stroke, influence the quality of life (QoL) in stroke survivors. A theoretical model based on Wilson and Cleary's model tested the specific hypotheses: 1) physical capability has an indirect effect on QoL mediated by BSE; 2) physical capability has an indirect effect on QoL by depressive symptoms; 3) stroke risk factors (hypertension/diabetes/gender) moderate the above relationship. Six hundred and seventy stroke survivors were enrolled from ten different hospitals in Yunnan province from 2019 to 2021. Patients' mental and physical function was assessed using the Brunnstrom recovery stage (BRS), mini-balance evaluation system test (Mini-BEST), Barthel index (BI), Activities-specific Balance Confidence scale (ABC), and Hamilton depression scale (HAM-D). The structural equation model (SEM) was used to test the moderated mediation model in Mplus 8.0 software. The model showed a good fit (RMSEA = 0.075, SRMR = 0.010). BSE significantly mediated the relationship between physical capability and QoL (β = 0.322, *p* = 0.002). Hypertension was found a significant moderator of all the direct paths from physical capability to QoL through depressive symptoms (**B** = 0.412, *p* = 0.015; **B** = 0.831, *p* = 0.020, respectively). This study provides a better insight into the relationship between physical capability and QoL via BSE in stroke survivors, which may help establish appropriate treatment for these individuals.

## Introduction

Stroke is a primary cause of long-term neurological disability, with more than 80 million stroke survivors globally in 2016 [1]. Compared to Europe, America, and Australia, Asia has the largest stroke burden, especially in China due to its large and aging population [2,3]. Stroke decreases the survivors' quality of life (QoL) because it can cause physical and cognitive impairments, emotional distress, and fatigue[4,5]. Additionally, it can also lead to social isolation and a loss of independence[6]. Therefore, it is a serious health problem worldwide, especially in Asia [7]. The reduction in QoL often accompanies the increased risk of lifestyle diseases, including hypertension, diabetes, and an array of physical dysfunctions. Thus,

**Data availability statement:** All relevant data are within the paper and its Supporting Information files.

**Funding:** The work was supported by the Major Science and Technology Project of Yunnan Province (Grant numbers[2018zf016]), Rehabilitation Clinical Medical Centre of Yunnan Province (Grant numbers[zx2019-04-02]), National Key Research and Development Program (Grant numbers[2018YFC2002301]), and Jiajie Expert Workstation of Yunnan Province (Grant numbers[2019IC034]), and Kunming Medical University Postgraduate Innovation Fund (Grant numbers[2022B11]). The funders had no role in study design, data collection and analysis, decision to publish, or preparation of the manuscript.

**Competing interests:** The authors have declared that no competing interests exist.

improving QoL can improve the overall health and physical ability of stroke survivors. Various factors, including physical [8] and psychological [9], affect the QoL of stroke survivors, which must be taken into account by rehabilitation professionals to improve the QoL.

A recent study showing a strong relationship between physical capability and QoL in stroke survivors has raised serious concerns [10]. Physical capability denotes one's capacity to completely undertake a task. Normally, it is evaluated based on the assessment of an individual's balance and motor functions, such as the MiniBest scale [11], and Brunnstrom stages of stroke recovery [12]. Earlier research has shown that better physical capability is an important predictor of stroke survivors' QoL [8]. Additionally, physical capability is associated with balance function [13], which may also pose risks to patients' QoL [14,15].

Balance impairment could lead stroke survivors to develop anxiety-related or depression-related distress behaviors such as fear of falling [16,17]. The fear of falling in stroke survivors frequently leads to activity and participation limitations as a consequence of poststroke physical disability [17]. Therefore, enhancing the balance confidence may improve patients' QoL [18]. Balance confidence can be assessed using the "Activities-specific Balance Confidence Scale (ABC)" questionnaire [19]. It is also called Balance self-efficacy (BSE), which means an individual's belief in his or her ability to accomplish a task [20]. Studies in the last few decades have produced fairly convergent results, indicating that higher BSE is linked to better QoL of stroke survivors [21–23]. French et al. proposed a theoretical model to explain the mediating role of BSE on the relationship between the patient's physical capability and participation [20]. Using the above theoretical model, we propose the first hypothesis, considering the mediation effect of BSE on the relationship between physical capability and QoL.

Additionally, poor QoL has been found to be strongly associated with severe depressive symptoms [24,25]. Evidence suggests that depressive symptoms after stroke harm motor outcomes [26]. A study reported that there is a negative relationship between depression and the perception of QoL [27]. Conversely, physical impairments contribute to post-stroke depressive disorders [28]. Thus, we propose the second hypothesis that the relationship between physical capabilities and QoL may also be mediated by depressive symptoms.

Besides, other risk factors such as hypertension, diabetes, and gender may further impact these relationships among physical capability, depressive symptoms, BSE, and QoL [29,30]. Hypertension is a leading risk factor for stroke in Asia [3,7], significantly increasing the risk of ADL disability [31]. Hypertension with other comorbid conditions lowers the patient's QoL and physical capability [32,33]. Interestingly, one study demonstrated that enhancing the self-efficacy of stroke survivors can help them improve blood pressure [34]. Concerning diabetes, ischemic stroke is a well-known macrovascular complication of diabetes. A study reported that diabetic stroke survivors have poorer outcomes than non-diabetic stroke survivors [35]. Notably, one study indicated gender association with health-related QoL [35]. Therefore, this study also examined the effect of these moderators (hypertension, diabetes, gender) on the relationships among physical capability, BSE, depressive symptoms, and QoL. Accordingly, here, we propose the third hypothesis that hypertension/diabetes/gender may act as moderators affecting the mediated relationship in stroke survivors [36].

To fully elucidate the complex relationships among factors that contribute to QoL in stroke, Wilson and Cleary's 1995 model (W-C model) is applied to link physical function (physical capability), general health perception (BSE, depressive symptoms), and QoL [37]. General health perception refers to self-reported health assessment [38]. A hypothetic and theoretic model of the expected moderated mediation is described in Fig 1. The model tests specific hypotheses: (1) physical capability has an indirect effect on QoL mediated by BSE; (2) physical capability has an indirect effect on QoL by depressive symptoms; (3) stroke risk factors (hypertension/diabetes/gender) moderate the above relationship.

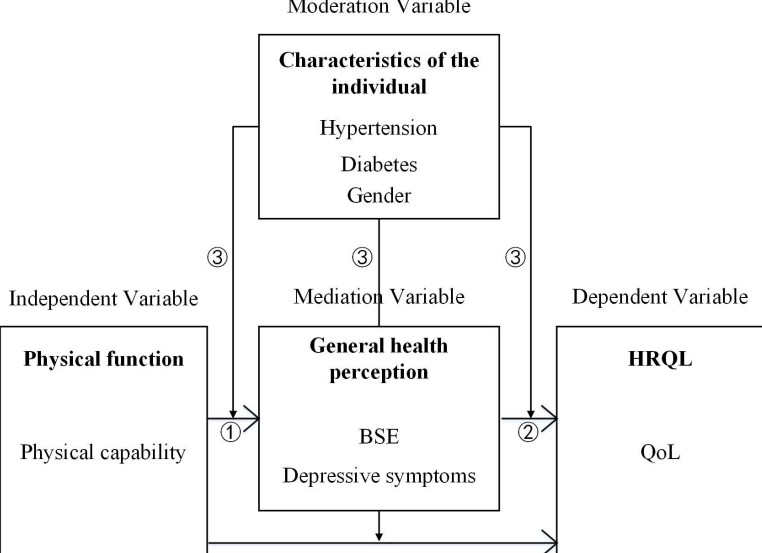

**Fig 1. The theoretical model is based on Wilson and Cleary's model.** ①: Hypothesis 1 of the current study; ②: Hypothesis 2 of the current study; ③: Hypothesis 3 of the current study. BSE: Balance Self-Efficacy; HRQL: Health-Related Quality of Life; QoL: Quality of Life. The arrows represent the hypothesized directional relationships between the variables in the model.

## Methods

### Study design

The research was a multicenter cross-sectional study conducted from April 2019 to April 2020. In total, 1,765 participants were recruited using the consecutive sampling method, consisting of inpatients from ten different rehabilitation institutions. Ethical approval was obtained from the local ethics committees (approval No. PJ-2019-04) and the research was performed according to the Declaration of Helsinki [39]. Written informed consent was obtained from all patients before participation.

### Participants

Patients' participation criteria were as follows: (I) age, > 18 years; (II) diagnosis of hemorrhagic or ischemic stroke for > 1 month; (III) stable vital signs, and (IV) signed informed consent. The participant exclusion criteria were as follows: (I) presence of any mental diagnosis before stroke according to the Diagnostic and Statistical Manual of Mental Disorders classification (i.e., major depressive disorder, bi-directional affective disorder, and obsessive-compulsive disorder, etc.); (II) malignant tumors, and/or; (III) dementia, loss of consciousness, or severe cardiopulmonary, liver, or kidney dysfunctions; (IV) pre-existing physical disability from any cause, such as fracture, osteoarthritis, etc. Out of the 1,765 participants, 245 did not meet the inclusion criteria, and 850 were excluded based on exclusion criteria. Finally, 670 participants were included in the structural equation model analysis.

### Procedure

The case data used in this cross-sectional study were obtained from stroke patients across ten rehabilitation facilities in Yunnan Province, China. Initially, ten hospitals were invited to

participate in a training program conducted every three months. Subsequently, two trained researchers visited various hospitals periodically to collect data and perform quality control [13]. After patients provided informed consent, data collectors conducted a standardized assessment, which included a neurological evaluation covering demographic information, clinical data, and both neurological and neuropsychological functional assessments. Assessment outcomes and patient information were entered into the electronic database of our hospital department (Department of Rehabilitation, the Second Affiliated Hospitals of Kunming Medical University).

## Measurement

Physical capability: The Brunnstrom recovery stage (BRS) [40] is conducted to evaluate post-stroke motor recovery in the affected limbs [12]. The evaluation consists of six levels, Stage 1 (flaccid paresis) to Stage 6 (approximately normal paresis). The total score is the summation of all scores of the limbs. A high score suggests a positive recovery. The Rasch reliability coefficient was above 0.90 [41]. Mini-BEST, a mini-balance evaluation system test [11], assesses the dynamic balance of the patient. The mini-BEST had high internal consistency and interrater reliability [42]. It includes 14 items, each rated from 0 (severe) to 2 (normal). The maximum score is 28 points. A higher score indicates better balance and postural control.

Quality of life (QoL): The Barthel index (BI) [43], a functionally independent tool, evaluates the patient's performance in daily life activities, with a total score ranging from 0 to 100. A high score indicates a better ability to live independently. BI demonstrates good validity and reliability [44].

Balance self-efficacy (BSE): ABC, an Activities-specific Balance Confidence scale, evaluates individuals' balance confidence while carrying out specific activities [45]. ABC shows excellent internal consistency and test-retest reliability [46]. This 11-point scale ranges from 0 (no confidence) to 100 (complete confidence). The total score denotes the average score.

Depressive symptoms: The Hamilton depression scale (HAM-D) is widely used to assess the severity of depression [47]. HAM-D demonstrates satisfactory reliability [48]. It included 17 items that were scored on a scale of 0 to 4. A higher score suggests a higher severity of depression.

## Statistics analysis

Descriptive statistics and bivariate correlation analysis were conducted with IBM SPSS 22.0. Normal data distribution was confirmed from the normal probability plots. Continuous data, with normal or near normal distribution, are expressed as mean ± SD; median and interquartile ranges $[P_{25}-P_{75}]$ are reported in case of nonnormal data distribution. Categorical data are expressed as rates and percentages. Pearson or Spearman correlation analyses were conducted to examine the correlations among BRS, Mini-BEST, QoL, BSE, and depressive symptoms.

The Mplus 8.0 program [49] was used to employ structural equation modeling (SEM) to test the hypothesized model. Some variables (<5%) had missing data. Missing data imputation uses a process of multiple imputations assuming data missing at random [50]. The maximum likelihood estimation method was applied to assess the structural equation model. The goodness-of-fit indices are widely used for structural equation models [51]. $\chi^2/df \leq 5$ (the ratio of $\chi^2$ to the degree of freedom) indicates an acceptable fitting of the data. The comparative five indexes (CFI) and the Tucker-Lewis index (TLI) of > 0.95 represent an excellent model fit. The SRMR (standardized root mean square residual) and RMSEA (root mean square error of approximation) indices of ≤ 0.08 indicate an acceptable fit of the model. $p < 0.05$ denotes the statistical significance of the data.

Additionally, a recommended bias-corrected bootstrapped method was employed to examine the mediators of BSE and depressive symptoms on the relationship between physical

capability and QoL and generate 95% bias-corrected confidence intervals (CIs) [52], with the procedure of 5,000 samples. A multi-group comparison test was employed to examine the indirect effects of physical capability and QoL differed between BSE and depressive symptoms. Also, the possible moderating effects of hypertension, diabetes, and gender among simple paths in the multiple-mediation model were examined. A statistically significant difference in the path coefficients between subgroups indicates the moderating effect of the path [36].

## Results

### Demographic characteristics

Six hundred and seventy participants (438 males, 232 females, mean age 61.4 ± 11.9 years) were involved in this study. The majority of participants (556, 83.0%) had an ischemic stroke, while hemorrhagic- and other strokes were reported in 101 (15.1%) and 13 (1.9%) patients respectively. Of the total number, 292 (43.6%) had a left-sided stroke (right hemiparesis), and 297 (44.3%) had a right-sided stroke (left hemiparesis). Patients with hypertension accounted for 68.5% (459), while patients with diabetes accounted for 11.3% (76).

### Correlations

The median score of the ABC was 88.13, whereas the mean assessment scales of mini-BEST, BRS, BI, and HAM-D are outlined in Table 1. We found significant correlations between BRS, Mini-BEST, QoL, BSE, and depressive symptoms ($p < 0.001$) (Table 2). BRS showed a positive correlation with Mini-BEST (r = 0.696), QoL (r = 0.633), and BSE (r = 0.644), but was negatively correlated with depressive symptoms (r = -0.438). Additionally, Mini-BEST showed a significantly positive correlation with BSE, and QoL (r = 0.789, r = 0.701), but a negative correlation with depressive symptoms (r = -0.492). BSE was positively correlated with QoL (r = 0.758) and negatively correlated with depressive symptoms (r = -0.496). QoL was negatively associated with depressive symptoms (r = -0.535).

### Fitting of the theoretical model

The SEM analysis showed that the study model fit indices indicated satisfactory fit ($\chi^2$ = 14.233, $\chi^2$/df = 4.744, $p$ = 0.002. RMSEA = 0.075, SRMR = 0.010, CFI = 0.994, TLI = 0.981).

### Path analysis

As shown in Fig 2, physical capability was the independent variable; BSE and depressive symptoms were the mediators and QoL was the dependent variable in the multiple mediation model. The standardized path coefficients, interpreted similarly to regression coefficients, are presented in Fig 2. All the pathways between physical capability and QoL were significant in the model. This model explained a total of 66.4% of the variance in QoL. Physical capability was a positive predictive factor of QoL ($\beta$ = 0.470, $p$ < 0.000). BSE contributed most to the total variance of QoL (71.8%), while Depressive symptoms accounted for 31.6% of the variance of QoL. The mediator of BSE positively predicted QoL ($\beta$ = 0.299, $p$ < 0.000). However, depressive symptoms were the negative predictor of QoL ($\beta$ = -0.122, $p$ < 0.000). Additionally, physical capability positively predicted self-efficacy ($\beta$ = 0.847, $p$ < 0.000) but was a negative of depressive symptoms ($\beta$ = -0.562, $p$ < 0.000) (Table 3).

### Mediation effect analysis

As shown in Table 4, we found significant total, direct, and indirect effects in the multiple mediation model. The specific indirect effect of physical capability on QoL via BSE was

**Table 1. Demographic and clinical information.**

| Variables | N (%)/mean±SD |
|---|---|
| **Age, mean (SD)** | 61.4 (11.9) |
| **Gender, N (%)** | |
| Male | 438 (65.4) |
| Female | 232 (34.6) |
| **Time since stroke, mean (SD)** | 143.8 (530.2) |
| **Type of stroke, N (%)** | |
| Ischemic | 556 (83.0) |
| Hemorrhagic | 101 (15.1) |
| Others | 13 (1.9) |
| **Paretic side, N (%)** | |
| Left | 292 (43.6) |
| Right | 297 (44.3) |
| Unknown | 81 (12.1) |
| **Dysfunction, N (%)** | |
| Limb functional disorder | 571 (85.2) |
| Speech dysfunction | 223 (33.3) |
| Cognitive disorder | 15 (2.2) |
| Balance disorder | 52 (7.8) |
| **Complication, N (%)** | |
| Hypertension | 459 (68.5) |
| Diabetes | 76 (11.3) |
| Cancer | 2 (0.3) |
| **Previous history, N (%)** | |
| Operation History | 6 (0.9) |
| Stroke History | 153 (22.8) |
| **Assessment scales, mean (SD)** | |
| Mini-BEST | 24.17 (7.247) |
| BRS | 15.18 (3.346) |
| ABC, Median (inter-quartile range) | 88.13 (64.38 95.63)[a] |
| Barthel index | 91.21 (13.740) |
| HAM-D | 3.67 (4.68) |

ABC: Activity-specific Balance Confidence; HAM-D: Hamilton Depression Scale; Mini-BEST: Mini-Balance Evaluation Systems Test; BRS: Brunnstrom; [a]: Median and inter-quartile range.

**Table 2. Correlations among variables.**

| | BRS | Mini-BEST | BSE | QoL | Depressive Symptoms | Mean | SD |
|---|---|---|---|---|---|---|---|
| **BRS** | 1 | | | | | 15.18 | 3.346 |
| **Mini-BEST** | 0.696[aa] | 1 | | | | 24.17 | 7.247 |
| **BSE** | 0.644[aab] | 0.789[aab] | 1 | | | 76.90 | 25.61 |
| **QoL** | 0.633[aa] | 0.701[aa] | 0.758[aa] | 1 | | 91.21 | 13.74 |
| **Depressive Symptoms** | -0.438[aa] | -0.492[aa] | -0.496[aa] | 0.535[aa] | 1 | 3.67 | 4.68 |

[aa]: $p < 0.001$; [b]: Spearman correlation.

BRS: Brunnstrom; BSE: Balance Self-efficacy; QoL: Quality of Life; Mini-BEST: Mini-Balance Evaluation Systems Test; SD: Standard Deviation.

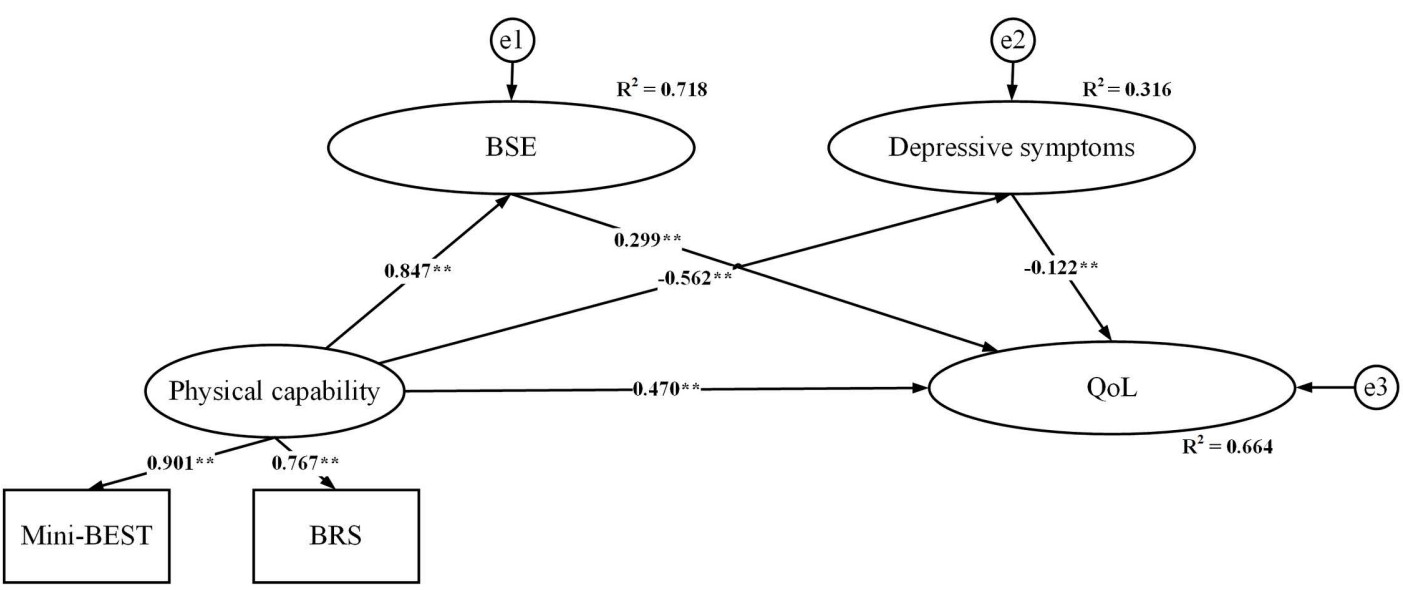

**Fig 2. Model fit ($\chi2$ = 14.233, $\chi^2$/df = 4.744, $p$ = 0.002. RMSEA = 0.075, SRMR = 0.010, TLI = 0.981, CFI = 0.994).** Mini-BEST: mini-Balance Evaluation Systems Test; BRS: Brunnstrom; BSE: Balance self-efficacy; QoL: Quality of Life. This model is consistent with hypotheses 1 and 2 of the theoretical model. BSE and Depressive symptoms are mediating variables between Physical capability and QoL.

**Table 3. Path analysis.**

| Variables | Point estimate | | Product of coefficients | | $p$ |
|---|---|---|---|---|---|
| | B | $\beta$ | SE | Z | |
| Physical capability---> QoL | 0.989 | 0.470 | 0.143 | 6.916 | 0.000[aa] |
| Physical capability---> BSE | 3.324 | 0.847 | 0.124 | 26.806 | 0.000[aa] |
| Physical capability---> Depressive symptoms | -0.403 | -0.562 | 0.026 | -15.500 | 0.000[aa] |
| BSE---> QoL | 0.16 | 0.299 | 0.031 | 5.161 | 0.000[aa] |
| Depressive symptoms---> QoL | -0.357 | -0.122 | 0.087 | -4.103 | 0.000[aa] |

B: Unstandardized coefficients; $\beta$: Standardized coefficients; SE: Standard error; QoL: Quality of Life; BSE: Balance Self-efficacy; [aa]: $p < 0.001$.

statistically significant ($\beta$ = 0.254, $p$ = 0.003). However, the other mediations of depressive symptoms were not statistically significant ($\beta$ = 0.069, $p$ = 0.136). These findings negated the hypothesis that depressive symptoms mediate the relationship between physical capability and QoL. Thus, BSE significantly mediates the association between physical capability and QoL ($\beta$ = 0.322, $p$ = 0.002).

## Moderation effect analysis

The analysis presented in Table 5 shows that hypertension significantly moderated the indirect effect of physical capability on QoL (**B** = -0.953, $p$ = 0.014). We examined the moderation effect using the pick-a-point approach to estimate the conditional direct effects of physical capability on QoL via BSE and depressive symptoms on QoL at different levels of hypertension. The results indicated that the interaction terms (physical capability × hypertension and depressive × hypertension) in the pathways from physical capability to QoL through depressive symptoms were significant (**B** = 0.412, $p$ = 0.015; **B** = 0.831, $p$ = 0.020, respectively). The plots of the interactions between physical capability, depressive symptoms, and QoL are

**Table 4. The total, direct, and indirect effects.**

| Effects | Variables | Point estimate | | Product of coefficients | | Bias-corrected percentile method | | p |
|---|---|---|---|---|---|---|---|---|
| | | B | β | SE | Z | Lower bounds | Upper bounds | |
| Total effect | Physical capability--->QoL | 4.242 | 0.793 | 0.265 | 16.108 | 3.756 | 4.805 | 0.000[aa] |
| Direct effect | Physical capability--->QoL | 2.517 | 0.470 | 0.610 | 4.125 | 1.325 | 3.720 | 0.000[aa] |
| Total indirect effect | Physical capability--->BSE--->QoL | 1.725 | 0.322 | 0.559 | 3.086 | 0.693 | 2.887 | 0.002[a] |
| | Physical capability--->Depressive symptoms--->QoL | | | | | | | |
| Specific indirect effect | Physical capability--->BSE--->QoL | 1.357 | 0.254 | 0.456 | 2.976 | 0.497 | 2.291 | 0.003[a] |
| Specific indirect effect | Physical capability--->Depressive symptoms--->QoL | 0.368 | 0.069 | 0.247 | 1.490 | -0.069 | 0.891 | 0.136 |

B: Unstandardized coefficients; β: Standardized coefficients; SE: Standard error; Z: Z-value; QoL: Quality of Life; BSE: Balance Self-efficacy; [a]: $p < 0.05$; [aa]: $p < 0.001$.

**Table 5. Moderated analysis.**

| Variables | Point estimate | Product of coefficients | | Bias-corrected percentile method | | p |
|---|---|---|---|---|---|---|
| | B | SE | Z | Lower bounds | Upper bounds | |
| Hypertension | -0.953 | 0.404 | -2.359 | -1.793 | -0.215 | 0.014[a] |
| Physical capability--->BSE | -0.956 | 0.925 | -1.034 | -2.765 | 0.844 | 0.301 |
| BSE--->QoL | -0.192 | 0.103 | -1.862 | -0.406 | 0.192 | 0.063 |
| Physical capability--->Depressive symptoms | 0.412 | 0.169 | 2.442 | 0.080 | 0.132 | 0.015[a] |
| Depressive symptoms--->QoL | 0.831 | 0.357 | 2.325 | 0.200 | 1.575 | 0.020[a] |
| Physical capability--->QoL | 2.897 | 1.239 | 2.339 | 0.574 | 5.462 | 0.019[a] |
| Diabetes | 0.943 | 1.660 | 0.568 | -1.455 | 3.683 | 0.570 |
| Gender | 0.022 | 0.476 | 0.046 | -0.835 | 1.015 | 0.961 |

B: Unstandardized coefficients; β: Standardized coefficients; SE: Standard error; Z: Z-value; QoL: Quality of Life; BSE: Balance Self-efficacy; [a]: $p < 0.05$.

presented in Fig 3. In all, hypertension significantly moderates the association between physical capability and QoL toward depressive symptoms, supporting our hypothesis. However, diabetes and gender showed no significant effect on the indirect path (**B** = 0.943, $p = 0.570$; **B** = 0.022, $p = 0.961$, respectively). Thus, these findings do not support the hypothesis that diabetes/gender moderates the aforementioned association.

## Discussion

This study primarily explored a multiple mediation model of the 'physical capability to QoL via BSE and depressive symptoms' pathway in a group of stroke survivors from China based on the Wilson and Cleary model. Our findings are consistent with a previous study, showing the significant mediation effect of BSE on the relationship between physical capability and QoL [20]. As hypothesized, the physical capability showed an indirect positive correlation with QoL through BSE; a higher level of physical capability indicated higher BSE, thereby improving the patient's QoL.

However, we found no mediation effect of depressive symptoms on the relationship between physical capability and QoL. According to French et al. [19], depressive symptoms moderate the relationships between physical capability, self-efficacy, and participation in stroke survivors. Although the hypothesis about depressive symptoms mediating the relationship between physical capability and QoL was not supported, all the paths in our model were significant. The present findings suggest that a high level of physical capability can predict a

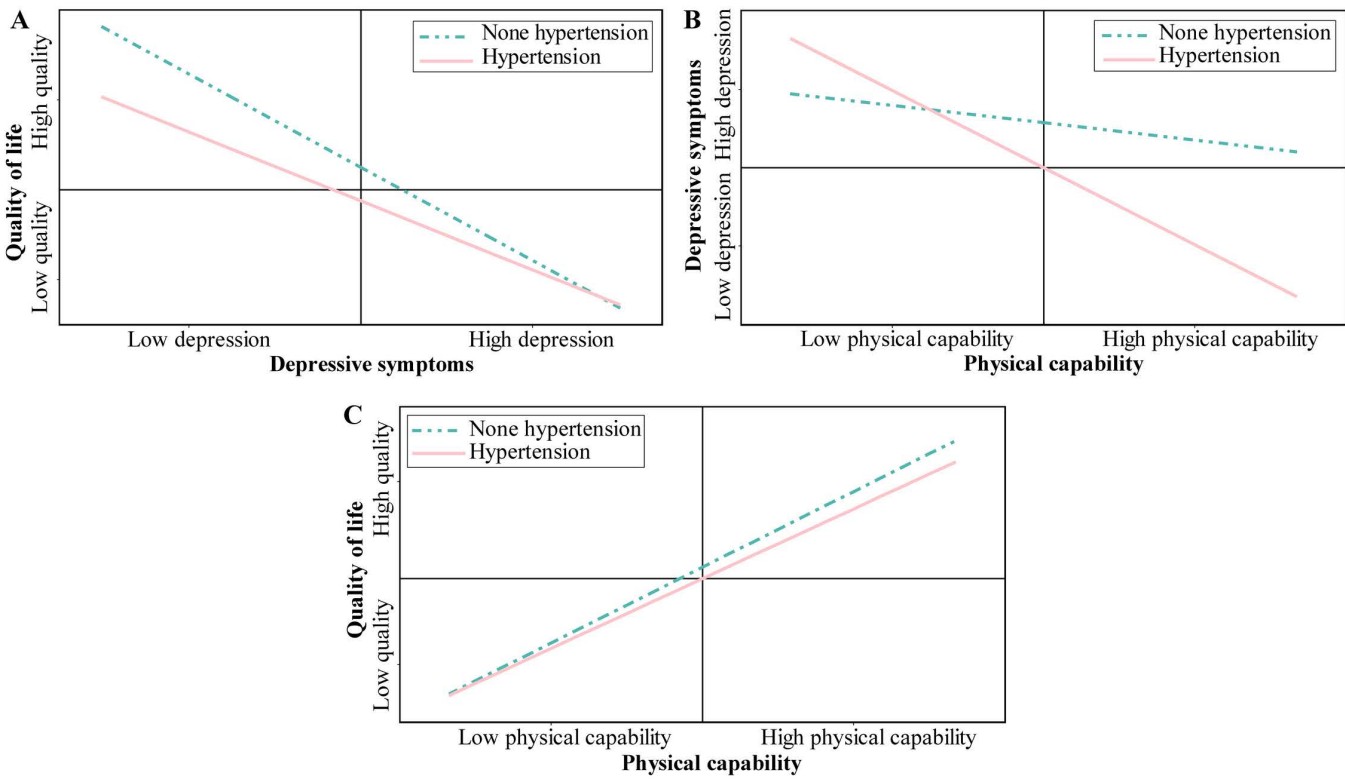

**Fig 3. Interactive effect of hypertension on the relationship between physical capability, QoL, and depressive symptoms.** This is consistent with the assumptions of Theoretical Model 3. Hypertension acts as a moderating variable between Physical capability and QoL.

lower severity of depressive symptoms and a better QoL, while a high severity of depressive symptoms predicts a lower QoL. This supports our hypothesis.

Our results also demonstrated that hypertension moderated the relationship between physical capability, depressive symptoms, and QoL in stroke survivors. The patients with hypertension had decreased physical capability, which in turn increased depressive symptoms. This result is consistent with Sok et al. showing a strong association between depressive symptoms and hypertension [35]. Accordingly, another study reported that depressed patients with hypertension had poor control of blood pressure [53]. The possibility exists that antidepressant therapy interferes with blood pressure control in persons with hypertension, but further research on this topic is warranted. Additionally, a recent study showed that increased walking (physical) activity lowers hypertension risk in postmenopausal women [54]. Our study also indicated that physical activity positively influences hypertension risk, even in a mixed-gender population. Conversely, patients with high blood pressure were found to be less active than those without hypertension. Furthermore, QoL negatively correlates with stroke risk factors such as hypertension [55, 56]. In general, hypertension is an important moderator of relationships between physical capability, depressive symptoms, and QoL.

However, there is a lack of literature examining the relationship between hypertension and BSE, which must be explored in both a qualitative and quantitative manner. Similarly, we found no evidence that hypertension moderated the 'physical capability to QoL via BSE' pathway. Although our findings did not show that diabetes and gender moderated the relationship between physical capability and QoL, this may be attributable to the relatively small number

of individuals with diabetes and females in this cohort. Therefore, further research is needed with a larger and more diverse sample.

The theoretical model in this study was built based on a previous model suggesting that BSE is a significant mediator and hypertension has a moderating effect on the 'physical capability to QoL through depressive symptoms' pathway. The mediating effect of BSE and the moderating effect of hypertension may help interpret the mechanisms of the relationship between physical capability and QoL. Our conceptual model facilitates the understanding of these associations to make targeted strategies promoting QoL and rehabilitation. Patients show improved balance function, which naturally enhances balance confidence, leading to a significant improvement in QoL [18]. Health professionals should incorporate interventions to enhance balance confidence and the control of blood pressure in their management of physical and mental rehabilitation.

## Limitations

There are some limitations to our research. First, we did not consider the complex nature of the QoL index, which may complicate relationships among the various missing data of clinical variables. Researchers should implement rigorous strategies to minimize missing data and place greater emphasis on exploring the intricacies of the QoL index in more detail, taking into account its multiple dimensions. Second, we did not take into account the impact of stroke subtypes, a possible confounding parameter in this study. Future research should consider stratifying stroke survivors by subtype, which would facilitate a more precise analysis of how different stroke types may influence the outcomes. Third, the cross-sectional study design restricts the interpretation and generalization of our results. An additional longitudinal study may help verify the conclusions of this study.

## Conclusion

This study provided a better insight into the relationship between physical capability and QoL via BSE in stroke survivors, which may help establish appropriate treatment for these individuals. Our results uncovered the underlying moderating effect of hypertension on relationships between physical capability, depressive symptoms, and QoL. An effective rehabilitation strategy can achieve concrete targets for stroke risk patients via BSE, improving their QoL [57].

## Supporting information

**S1 File. Data - revision.**
(XLSX)

**S2 File. Mplus_moderation.**
(XLSX)

## Acknowledgment

The authors would like to thank all the participants in the study and MJEditor (www.mjeditor.com) for their linguistic assistance during the preparation of this manuscript.

## Author contributions

**Conceptualization:** Xue Yang, Hongmei Zhang, Qian Liu, Yihuan Lu.

**Data curation:** Xue Yang, Qian Liu, Yihuan Lu.

**Investigation:** Yihuan Lu.

**Methodology:** Xue Yang, Hongmei Zhang, Qian Liu, Yihuan Lu.

**Project administration:** Liqing Yao.

**Supervision:** Liqing Yao.

**Writing – original draft:** Xue Yang, Hongmei Zhang.

**Writing – review & editing:** Xue Yang, Hongmei Zhang, Qian Liu.

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
