## [Decision Letter · Decision Letter 0]

3 Dec 2024

PONE-D-24-45706Mediation and moderation analysis of the association between physical capability and quality of life among stroke patientsPLOS ONE

Dear Dr. Yao,

Thank you for submitting your manuscript to PLOS ONE. After careful consideration, we feel that it has merit but does not fully meet PLOS ONE’s publication criteria as it currently stands. Therefore, we invite you to submit a revised version of the manuscript that addresses the points raised during the review process.

We look forward to receiving your revised manuscript.

Kind regards,

Diphale Joyce Mothabeng, PhD

Academic Editor

PLOS ONE

Journal Requirements:

“The work was supported by the Major Science and Technology Project of Yunnan Province (Grant numbers[2018zf016]), Rehabilitaion Clinical Medical Centre of Yunnan Province (Grant numbers[zx2019-04-02]), National Key Research and Development Program (Grant numbers[2018YFC2002301]), and Jiajie Expert Workstation of Yunnan Province (Grant numbers[2019IC034]), and Kunming Medical University Postgraduate Innovation Fund (Grant numbers[2022B11]).”

3. Please note that funding information should not appear in the Acknowledgments section or other areas of your manuscript. We will only publish funding information present in the Funding Statement section of the online submission form. Please remove any funding-related text from the manuscript. 

4. We note that your Data Availability Statement is currently as follows: 

“All relevant data are within the manuscript and its Supporting Information files.”

7. Please include a separate caption for each figure in your manuscript.

**Additional Editor Comments:**

Thank you for the submission. Kindly attend to the reviewers comments that the journal will provide you with.

Reviewers' comments:

Reviewer's Responses to Questions

**Comments to the Author**

1. Is the manuscript technically sound, and do the data support the conclusions?

Reviewer #1: Yes

Reviewer #2: Yes

2. Has the statistical analysis been performed appropriately and rigorously? 

Reviewer #1: Yes

Reviewer #2: Yes

3. Have the authors made all data underlying the findings in their manuscript fully available?

Reviewer #1: Yes

Reviewer #2: Yes

4. Is the manuscript presented in an intelligible fashion and written in standard English?

Reviewer #1: Yes

Reviewer #2: No

5. Review Comments to the Author

Reviewer #1: The manuscript was well written. There are a few minor corrections to be made. These are in the track-changed document attached.

I t will be great to discuss how stroke affects the qukity of life of stroke survivors. Also, the results on BRS, MiniBest, QoL, BSE abd depresion has to be pesented first before the correlations.

Reviewer #2: This is a very interesting article, in that it provides sound scientific evidence of suspicions that we all had regarding stroke patients. The researchers are commended for the work that was done.

I only have minor suggestions to improve the readability of the information provided and you will see that the majority of the comments are language/editing related - it might be advisable to consider using a language editor to review the paper as well:

1) Throughout the document, 'capacity' and 'capability' are used interchangeably. Although these words do have the same meaning, it will be easier to read and process the information if the same term is used throughout (I would recommend using 'capability' as indicated in the title). If the researchers did not intend to use the words as synonyms, it would be helpful to include a definition or description of the difference between the terms in the introduction section.

2) In the abstract, under the results heading there is a ' before physical capability (last sentence of the results section). Please either remove the ' or complete the statement by adding a ' after depressive symptoms.

3) In the introduction section, the first sentence is a bit difficult to read because of the punctuations used, my suggestion would be to change the sentence to: Stroke is a primary cause of long-term neurological disability, with >80 million stroke survivors globally in 2016. (so remove the ; and change worldwide to globally to enhance the flow of the sentence).

4) In the second paragraph in the introduction (balance), the second sentence is a bit unclear (i.e., is the sedentary lifestyle a result of the stroke or as a result of the fear of falling)? Please rephrase for clarity.

5) In the balance paragraph, in the sentence after the ABC questionnaire, please change it to it is (i.e., it is also called)

6) Please revise the sentence in the balance paragraph (after reference nr 17) for ease of reading, my suggestion would be: Studies in the last few decades have produced fairly convergent results, indicating that higher balance self-efficacy is linked to better QOL of stroke patients.

7) In the sentence after reference 21, please remove the word 'here'.

8) In the first sentence on page 11 (after reference nr 24), please change to 'One research study'. In the following sentence, please change contributes to contribute.

9) In the introductory paragraph, regarding hypertension, please elaborate a bit more in terms of what you mean by 'significantly affecting daily lives' - is this activities of daily living?

10) At the conclusion of the introduction section, the information is a bit confusing (although I do see that you merged hypothesis one and two and reported it as one) - but this can be very confusing. I would recommend that you either adhere to the two hypothesis (as indicated at the end of the introduction section throughout) or that you write out all three (without merging the described hypothesis 1 and 2). Which ever you decide, please remember to adapt the abstract accordingly.

11) In figure 1, it would be nice to add either colour or a bit more information, so that it is clear which factors are hypothesised to be mediators and which factors are moderators (to align more clearly with the written information).

12) Regarding the exclusion criteria, please be a bit more specific regarding 'several mental disorders' - are these disorders classified on the DSM and how many were considered to be too many (for example, if a patient had anxiety and depression were they excluded, or were more than 2 diagnoses required for exclusion?). Were previous disability excluded as well? For example, if the patient had an amputation / severe orthopaedic condition? (as these factors may influence the results of the study)

13) Figure 2 does not really indicate any additional information from what was provided in the text and the researchers can consider omitting it from the manuscript.

14) In the Procedure section (under the Methods heading) I referred to the referenced paper regarding quality controls, but it is not clear what the researchers are referring to - i.e., what quality controls were used? (This information is from the referenced study: Subsequently, 10 hospitals were invited to participate in relevant training programs every 3 months, and two trained investigators made irregular visits to different hospitals to collect information and conduct quality control. - is this what the researchers are referring to? Because it is not a lot of information, it might be useful to write it out in the current study (under review).

15) In the results section, please revise the second sentence, I suspect it is supposed to read: The majority of participants had an ischemic stroke, while hemorrhagic- and other strokes were reported in 101 and 13 patients respectively.

16) Please clarify whether left stroke and right stroke refers to left/right hemiplegia/paresis or left or right cerebral involvement.

17) Just for noting - I found the path analysis very interesting and Figure 3 really assisted with interpreting and understanding all the information (which can sometimes be a bit overwhelming)

18) I think your interpretation of the diabetes/gender effect may be a bit harsh, i.e., that your hypothesis was rejected. I would recommend that you add that although the results from your study didn't support your hypothesis, the smaller (significantly so?) number of females (compared to males) and the small number of patients with diabetes may have influenced the results.

19) In the discussion section, in the first sentence of the second paragraph (however, we found no...) please elaborate the sentence a bit more so that the 'relationship' you are referring to is clear.

20) Discussion section: When using in-text references, you don't have to include the initials (so it will become Sok et al.)

21) Please reference the sentence on page 17, i.e., 'it could be that antidepressant therapy interferes with blood pressure control in hypertension patients' - alternatively, if this is not a proven fact, indicate it (for example, the possibility exists that antidepressant therapy interferes with blood pressure control in hypertensive patients, but further research on the topic is warranted).

22) The sentence directly after the one referred to in comment 21 needs revision - so that the link between postmenopausal women and the study becomes clear (for example: additionally, a recent study showed that increased walking (physical) activity lowers hypertension risk in postmenopausal women [52]. Our study also indicated that physical activity positively influences hypertension risk, even in a mixed-gender population.

23) The final sentence of the discussion needs to be preceded by a sentence regarding balance so that the linkage becomes clear.

24) Throughout the discussion section OR (alternatively) in the limitations section, the researchers should make some recommendations for future research (either to say how limitations can be minimised in future research, or recommendation for future research based on the findings of the study.

25) Please ensure the correct tense is used in the conclusion paragraph (the study has been conducted, so the section should be in the past tense).

6. PLOS authors have the option to publish the peer review history of their article (what does this mean? ). If published, this will include your full peer review and any attached files.

**Do you want your identity to be public for this peer review?** For information about this choice, including consent withdrawal, please see our Privacy Policy .

Reviewer #1: **Yes: ** Tawwagidu Mohammed

Reviewer #2: **Yes: ** Maria Elizabeth Cochrane

---

## [Author Response · Author response to Decision Letter 1]

3 Jan 2025

Dear editors and reviewers:

Re: Manuscript ID: PONE-D-24-45706 and Title: Mediation and moderation analysis of the association between physical capability and quality of life among stroke patients

Thank you for your letter and the reviewers’ comments concerning our manuscript entitled “PONE-D-24-45706” (ID). Those comments are valuable and very helpful. We have read through the comments carefully and have made corrections. Based on the instructions provided in your letter, we uploaded the file of the revised manuscript. Revisions in the main text are highlighted in yellow as additions or modifications. The responses to the reviewer's comments are marked in yellow and presented following.

We would love to thank you for allowing us to resubmit a revised copy of the manuscript and we highly appreciate your time and consideration.

Sincerely.

Liqing-Yao

Editorial opinion:

1) *If applicable, we recommend that you deposit your laboratory protocols in protocols.io to enhance the reproducibility of your results. Protocols.io assigns your protocol its identifier (DOI) so that it can be cited independently in the future. For instructions see: https://journals.plos.org/plosone/s/submission-guidelines#loc-laboratory-protocols. Additionally, PLOS ONE offers an option for publishing peer-reviewed Lab Protocol articles, which describe protocols hosted on protocols.io. Read more information on sharing protocols at https://plos.org/protocols?utm_medium=editorial-email&utm_source=authorletters&utm_campaign=protocols.*

Response: Thanks for the editorial suggestions.

Journal Requirements:

1)*Please ensure that your manuscript meets PLOS ONE's style requirements, including those for file naming. The PLOS ONE style templates can be found at

Response: Thanks to the editorial advice, I have downloaded the PLOS ONE style template and have modified it in this way, e.g. the title page in the manuscript as well.

2)* Thank you for stating the following financial disclosure:

“The work was supported by the Major Science and Technology Project of Yunnan Province (Grant numbers[2018zf016]), Rehabilitation Clinical Medical Centre of Yunnan Province (Grant numbers[zx2019-04-02]), National Key Research and Development Program (Grant numbers[2018YFC2002301]), and Jiajie Expert Workstation of Yunnan Province (Grant numbers[2019IC034]), and Kunming Medical University Postgraduate Innovation Fund (Grant numbers[2022B11]).”

Response: We have clarified the role of the funder in the study and have removed the funding section from the manuscript and added a funding section to the cover letter. This corresponds to lines 12 to 19 of the cover letter.

3)*Please note that funding information should not appear in the Acknowledgments section or other areas of your manuscript. We will only publish funding information present in the Funding Statement section of the online submission form. Please remove any funding-related text from the manuscript.

Response: We have removed any fund-related text from the manuscripts. We have added a funding section to the cover letter. This corresponds to lines 12 to 19 of the cover letter. This corresponds to lines 12 to 19 of the cover letter.

4)* We note that your Data Availability Statement is currently as follows:

“All relevant data are within the manuscript and its Supporting Information files.”

Please confirm at this time whether or not your submission contains all the raw data required to replicate the results of your study. Authors must share the “minimal data set” for their submission. PLOS defines the minimal data set to consist of the data required to replicate all study findings reported in the article, as well as related metadata and methods (https://journals.plos.org/plosone/s/data-availability#loc-minimal-data-set-definition).

- The values behind the means, standard deviations, and other measures reported;

Response: Thanks to the suggestion of the editorial team, we have uploaded the supporting data files named “data.xlsx” and “Mplus_moderation.xlsx”. The “data.xlsx” file is the full information of the 670 patients we included, while “Mplus_moderation.xlsx” is used to draw the patient information for structural equation modeling.

5)*PLOS requires an ORCID iD for the corresponding author in Editorial Manager on papers submitted after December 6th, 2016. Please ensure that you have an ORCID iD and that it is validated in Editorial Manager. To do this, go to ‘Update my Information’ (in the upper left-hand corner of the main menu), and click on the Fetch/Validate link next to the ORCID field. This will take you to the ORCID site and allow you to create a new ID or authenticate a pre-existing iD in Editorial Manager.

Response: Thanks to the editorial board's suggestion, we submitted this manuscript using the corresponding author's ORCID iD login to the PLOS ONE website.

6)*Your ethics statement should only appear in the Methods section of your manuscript. If your ethics statement is written in any section besides the Methods, please delete it from any other section.

Response: I have removed any part of the ethics statement written outside of the method.

7)* Please include a separate caption for each figure in your manuscript.

Response: I have captioned each figure separately in the manuscript as shown in Figure 1.

8)* Please review your reference list to ensure that it is complete and correct. If you have cited papers that have been retracted, please include the rationale for doing so in the manuscript text, or remove these references and replace them with relevant current references. Any changes to the reference list should be mentioned in the rebuttal letter that accompanies your revised manuscript. If you need to cite a retracted article, indicate the article’s retracted status in the References list and also include a citation and full reference for the retraction notice.

Response: I have checked the reference list.

Reviewer #1:

1)*The manuscript was well written. There are a few minor corrections to be made. These are in the track-changed document attached.

It will be great to discuss how stroke affects the quality of life of stroke survivors. Also, the results on BRS, MiniBest, QoL, BSE, and depression have to be presented first before the correlations.

Response: We thank the reviewers for their suggestions and we have revised this section.

Reviewer #2:

1)*Throughout the document, 'capacity' and 'capability' are used interchangeably. Although these words do have the same meaning, it will be easier to read and process the information if the same term is used throughout (I would recommend using 'capability' as indicated in the title). If the researchers did not intend to use the words as synonyms, it would be helpful to include a definition or description of the difference between the terms in the introduction section.

Response: I thank the reviewers for their careful suggestions. I have replaced “physical capacity” with “physical capability” throughout the manuscript and highlighted the changes in yellow.

2)*In the abstract, under the results heading there is a ' before physical capability (last sentence of the results section). Please either remove the ' or complete the statement by adding an ' after depressive symptoms.

Response:

I would like to thank the reviewers for their careful reading. I have revised and highlighted the text in yellow. This corresponds to lines 44 to 46 of the manuscript.

3)*In the introduction section, the first sentence is a bit difficult to read because of the punctuation used, my suggestion would be to change the sentence to: Stroke is a primary cause of long-term neurological disability, with >80 million stroke survivors globally in 2016. (so remove the , and change worldwide to globally to enhance the flow of the sentence).

Response: Thank you to the reviewers for their honest advice. I have revised and highlighted the text in yellow. This corresponds to lines 50 to 51 of the manuscript.

4)*In the second paragraph in the introduction (balance), the second sentence is a bit unclear (i.e., is the sedentary lifestyle a result of the stroke or as a result of the fear of falling)? Please rephrase for clarity.

Response: I would like to thank the reviewers for their careful reading. I have revised and highlighted the text in yellow. This corresponds to lines 71 to 73 of the manuscript.

5� *In the balance paragraph, in the sentence after the ABC questionnaire, please change it to it is (i.e., it is also called)

Response: I would like to thank the reviewers for their careful reading. I have revised and highlighted the text in yellow. This corresponds to lines 75 to 76 of the manuscript.

6)*Please revise the sentence in the balance paragraph (after reference nr 17) for ease of reading, my suggestion would be: Studies in the last few decades have produced fairly convergent results, indicating that higher balance self-efficacy is linked to better QOL of stroke patients.

Response: I would like to thank the reviewers for their careful reading. I have revised and highlighted the text in yellow. This corresponds to lines 77 to 79 of the manuscript.

7)*In the sentence after reference 21, please remove the word 'here'.

Response: I would like to thank the reviewers for their careful reading. I have revised and highlighted the text in yellow. This corresponds to lines 81 to 82 of the manuscript.

8)*In the first sentence on page 11 (after reference nr 24), please change to 'One research study'. In the following sentence, please change contributes to contribute.

Response: I would like to thank the reviewers for their careful reading. I have revised and highlighted the text in yellow. This corresponds to line 87 of the manuscript.

9)*In the introductory paragraph, regarding hypertension, please elaborate a bit more in terms of what you mean by 'significantly affecting daily lives' - are these activities of daily living?

Response: We thank the reviewers for their careful suggestions, and we have revised them here. This corresponds to lines 92 to 93 of the manuscript.

10)*At the conclusion of the introduction section, the information is a bit confusing (although I do see that you merged hypothesis one and two and reported it as one) - but this can be very confusing. I would recommend that you either adhere to the two hypotheses (as indicated at the end of the introduction section throughout) or that you write out all three (without merging the described hypotheses 1 and 2). Whichever you decide, please remember to adapt the abstract accordingly.

Response: We thank the reviewers for their careful suggestions, and we have revised them here. We have added hypothesis 3. This corresponds to lines 109 to 112 of the manuscript.

11)*In Figure 1, it would be nice to add either color or a bit more information, so that it is clear which factors are hypothesized to be mediators and which factors are moderators (to align more clearly with the written information).

Response: We thank the reviewers for their careful suggestions and we have revised Figure 1. This corresponds to lines 113 to 118 of the manuscript.

12)*Regarding the exclusion criteria, please be a bit more specific regarding 'several mental disorders' - are these disorders classified on the DSM, and how many were considered to be too many (for example, if a patient had anxiety and depression were they excluded, or were more than 2 diagnoses required for exclusion?). Were previous disability excluded as well? For example, if the patient had an amputation / severe orthopedic condition? (as these factors may influence the results of the study)

Response: I thank the reviewers for their honest advice. I have revised the exclusion criteria and highlighted the text in yellow. We refined the inclusion-exclusion criteria. This corresponds to lines 130 to 136 of the manuscript.

13)*Figure 2 does not indicate any additional information from what was provided in the text and the researchers can consider omitting it from the manuscript.

Response: Thank you to the reviewers for their suggestions. Since we have described the participant inclusion process in the Participants section of the methodology, the original Figure 2 has been removed.

14)*In the Procedure section (under the Methods heading) I referred to the referenced paper regarding quality controls, but it is not clear what the researchers are referring to - i.e., what quality controls were used? (This information is from the referenced study: Subsequently, 10 hospitals were invited to participate in relevant training programs every 3 months, and two trained investigators made irregular visits to different hospitals to collect information and conduct quality control. - is this what the researchers are referring to? Because it is not a lot of information, it might be useful to write it out in the current study (under review).

Response: I would like to thank the reviewers for their careful reading. We describe the data collection quality control process in detail. I have revised and highlighted the text in yellow. This corresponds to lines 140 to 147 of the manuscript.

15)*In the results section, please revise the second sentence, I suspect it is supposed to read: The majority of participants had an ischemic stroke, while hemorrhagic- and other strokes were reported in 101 and 13 patients respectively.

Response: I would like to thank the reviewers for their careful reading. I have revised and highlighted the text in yellow. This corresponds to lines 205 to 208 of the manuscript.

16)*Please clarify whether left stroke and right stroke refer to left/right hemiplegia/paresis or left or right cerebral involvement.

Response: Thank you to the reviewers for their suggestions. Left/right hemiparesis in the manuscript implies left/right paralysis or right/left stroke. This corresponds to lines 207 to 208 of the manuscript.

17)*Just for noting - I found the path analysis very interesting and Figure 3 assisted with interpreting and understanding all the information (which can sometimes be a bit overwhelming)

Response: Thank you to the reviewers for their suggestions. We have revised this figure and added a note to explain it. This corresponds to lines 247 to 253 of the manuscript.

18)*I think your interpretation of the diabetes/gender effect may be a bit harsh, i.e., that your hypothesis

---

## [Decision Letter · Decision Letter 1]

23 Jan 2025

PONE-D-24-45706R1Mediation and moderation analysis of the association between physical capability and quality of life among stroke patientsPLOS ONE

Dear Dr. Yao,

Thank you for submitting your manuscript to PLOS ONE. After careful consideration, we feel that it has merit but does not fully meet PLOS ONE’s publication criteria as it currently stands. Therefore, we invite you to submit a revised version of the manuscript that addresses the points raised during the review process.

Your manuscript, entitled "*Mediation and moderation analysis of the association between physical capability and quality of life among stroke patients* ", has been reviewed. Your efforts to revise the manuscript are appreciated. However, the peer review process continues because Reviewer 1 has a few additional comments that the author should address. Please find the reviewer's comments below.

We look forward to receiving your revised manuscript.

Kind regards,

Tanja Grubić Kezele, Ph.D., M.D.

Academic Editor

PLOS ONE

Journal Requirements:

Reviewers' comments:

Reviewer's Responses to Questions

**Comments to the Author**

1. If the authors have adequately addressed your comments raised in a previous round of review and you feel that this manuscript is now acceptable for publication, you may indicate that here to bypass the “Comments to the Author” section, enter your conflict of interest statement in the “Confidential to Editor” section, and submit your "Accept" recommendation.

Reviewer #1: All comments have been addressed

Reviewer #2: All comments have been addressed

2. Is the manuscript technically sound, and do the data support the conclusions?

Reviewer #1: Yes

Reviewer #2: Yes

3. Has the statistical analysis been performed appropriately and rigorously? 

Reviewer #1: Yes

Reviewer #2: Yes

4. Have the authors made all data underlying the findings in their manuscript fully available?

Reviewer #1: Yes

Reviewer #2: Yes

5. Is the manuscript presented in an intelligible fashion and written in standard English?

Reviewer #1: Yes

Reviewer #2: Yes

6. Review Comments to the Author

Reviewer #1: Well done on the manuscript. There are a few comments attached to address to further improve the work.

Reviewer #2: Thank you very much for addressing the comments thoroughly. The manuscript is ready for publication.

7. PLOS authors have the option to publish the peer review history of their article (what does this mean? ). If published, this will include your full peer review and any attached files.

**Do you want your identity to be public for this peer review?** For information about this choice, including consent withdrawal, please see our Privacy Policy .

Reviewer #1: **Yes: ** Tawagidu Mohammed

Reviewer #2: No

---

## [Author Response · Author response to Decision Letter 2]

12 Feb 2025

Dear editors and reviewers:

Re: Manuscript ID: PONE-D-24-45706R1 and Title: Mediation and moderation analysis of the association between physical capability and quality of life among stroke patients

Thank you for your letter and the reviewers’ comments concerning our manuscript entitled “PONE-D-24-45706R1” (ID). Those comments are valuable and very helpful. We have read through the comments carefully and have made corrections. Based on the instructions provided in your letter, we uploaded the file of the revised manuscript. Revisions in the main text are highlighted in yellow as additions or modifications. The responses to the reviewer's comments are marked in yellow and presented following.

We would love to thank you for allowing us to resubmit a revised copy of the manuscript and we highly appreciate your time and consideration.

Sincerely.

Liqing-Yao

Reviewer #1:

1)* There is the use of ‘stroke patients’ and ‘stroke survivors’ interchangeably. I think you would rather use ‘stroke survivors’. Kindly change all the stroke patients in the writing to survivors.

Response: I thank the reviewers for their honest advice. We have changed “stroke patients” to “stroke survivors” throughout the manuscript. I have made changes to the text and highlighted them in yellow, some of which are shown below.

2)* In the results section, there was a reference to fig1 but I couldn’t find fig1 in the results. Also, ensure that all the values in table 4 are presented such that the numbers (eg. 0.000) are in line instead of some coming under others. Therefore increase the width of the cells.

Response: Thank you to the reviewers for their constructive comments. I have changed the description in the results section. And I have revised Table 4 to make the table presentation more straightforward and aesthetically pleasing.

3)* There are also a few corrections in track changes.

Response: I thank the reviewers for their careful suggestions. We have revised all the marked places throughout the manuscript, some of which are shown below.

---

## [Decision Letter · Decision Letter 2]

26 Feb 2025

Mediation and moderation analysis of the association between physical capability and quality of life among stroke patients

PONE-D-24-45706R2

Dear Dr. Yao,

We’re pleased to inform you that your manuscript has been judged scientifically suitable for publication and will be formally accepted for publication once it meets all outstanding technical requirements.

Kind regards,

Tanja Grubić Kezele, Ph.D., M.D.

Academic Editor

PLOS ONE

Additional Editor Comments (optional):

Reviewers' comments:

Reviewer's Responses to Questions

**Comments to the Author**

1. If the authors have adequately addressed your comments raised in a previous round of review and you feel that this manuscript is now acceptable for publication, you may indicate that here to bypass the “Comments to the Author” section, enter your conflict of interest statement in the “Confidential to Editor” section, and submit your "Accept" recommendation.

Reviewer #1: All comments have been addressed

2. Is the manuscript technically sound, and do the data support the conclusions?

Reviewer #1: Yes

3. Has the statistical analysis been performed appropriately and rigorously? 

Reviewer #1: Yes

4. Have the authors made all data underlying the findings in their manuscript fully available?

Reviewer #1: Yes

5. Is the manuscript presented in an intelligible fashion and written in standard English?

Reviewer #1: Yes

6. Review Comments to the Author

Reviewer #1: Well done on improving the manuscript. All comments raised in the previous review have been adequately addressed

7. PLOS authors have the option to publish the peer review history of their article (what does this mean? ). If published, this will include your full peer review and any attached files.

**Do you want your identity to be public for this peer review?** For information about this choice, including consent withdrawal, please see our Privacy Policy .

Reviewer #1: **Yes: ** Tawagidu Mohammed

---

## [Editor Report · Acceptance letter]

PONE-D-24-45706R2

PLOS ONE

Dear Dr. Yao,

I'm pleased to inform you that your manuscript has been deemed suitable for publication in PLOS ONE. Congratulations! Your manuscript is now being handed over to our production team.

Kind regards,

on behalf of

Prof. dr. Tanja Grubić Kezele

Academic Editor

PLOS ONE